# Modulation of the Tumor Microenvironment with Trastuzumab Enables Radiosensitization in HER2+ Breast Cancer

**DOI:** 10.3390/cancers14041015

**Published:** 2022-02-17

**Authors:** Patrick N. Song, Ameer Mansur, Yun Lu, Deborah Della Manna, Andrew Burns, Sharon Samuel, Katherine Heinzman, Suzanne E. Lapi, Eddy S. Yang, Anna G. Sorace

**Affiliations:** 1Department of Radiology, The University of Alabama at Birmingham, Birmingham, AL 35294, USA; psong@uabmc.edu (P.N.S.); yunlu@uab.edu (Y.L.); sharonsamuel@uabmc.edu (S.S.); lapi@uab.edu (S.E.L.); 2Graduate Biomedical Sciences, The University of Alabama at Birmingham, Birmingham, AL 35294, USA; 3Department of Biomedical Engineering, The University of Alabama at Birmingham, Birmingham, AL 35294, USA; amansur@uab.edu (A.M.); acburns@uab.edu (A.B.); kheinz@uab.edu (K.H.); 4Department of Radiation Oncology, The University of Alabama at Birmingham, Birmingham, AL 35294, USA; dmanna@uabmc.edu (D.D.M.); shyang@uabmc.edu (E.S.Y.); 5O’Neal Comprehensive Cancer Center, The University of Alabama at Birmingham, Birmingham, AL 35294, USA

**Keywords:** PDX, BT474, MDA-MB-361, BCM 3472, trastuzumab, [^18^F]-FMISO, PET, synergy

## Abstract

**Simple Summary:**

Trastuzumab and radiation are used clinically to treat HER2-overexpressing breast cancers; however, the mechanistic synergy of anti-HER2 and radiation therapy has not been investigated. In this study, we identify that a subtherapeutic dose of trastuzumab sensitizes the tumor microenvironment to fractionated radiation. This results in longitudinal sustained response by triggering a state of innate immune activation through reduced DNA damage repair and increased tumor oxygenation. As positron emission tomography imaging can be used to longitudinally evaluate changes in tumor hypoxia, synergy of combination therapies is the result of both cellular and molecular changes in the tumor microenvironment.

**Abstract:**

DNA damage repair and tumor hypoxia contribute to intratumoral cellular and molecular heterogeneity and affect radiation response. The goal of this study is to investigate anti-HER2-induced radiosensitization of the tumor microenvironment to enhance fractionated radiotherapy in models of HER2+ breast cancer. This is monitored through in vitro and in vivo studies of phosphorylated γ-H2AX, [^18^F]-fluoromisonidazole (FMISO)-PET, and transcriptomic analysis. In vitro, HER2+ breast cancer cell lines were treated with trastuzumab prior to radiation and DNA double-strand breaks (DSB) were quantified. In vivo, HER2+ human cell line or patient-derived xenograft models were treated with trastuzumab, fractionated radiation, or a combination and monitored longitudinally with [^18^F]-FMISO-PET. In vitro DSB analysis revealed that trastuzumab administered prior to fractionated radiation increased DSB. In vivo, trastuzumab prior to fractionated radiation significantly reduced hypoxia, as detected through decreased [^18^F]-FMISO SUV, synergistically improving long-term tumor response. Significant changes in IL-2, IFN-gamma, and THBS-4 were observed in combination-treated tumors. Trastuzumab prior to fractionated radiation synergistically increases radiotherapy in vitro and in vivo in HER2+ breast cancer which is independent of anti-HER2 response alone. Modulation of the tumor microenvironment, through increased tumor oxygenation and decreased DNA damage response, can be translated to other cancers with first-line radiation therapy.

## 1. Introduction

Tumor hypoxia is an integral component of the tumor microenvironment that impacts tumor aggressiveness, metastatic potential, and response to therapy [1]. Radiation therapy is one of many treatments utilized as standard of care to treat human epidermal growth factor receptor 2 (HER2+) breast cancer in the primary and metastatic settings, which is defined by the amplification of the HER2 gene [2]. Radiation therapy converts oxygen present in tissue into reactive oxygen species (ROS), which will damage the DNA and cause genomic instability in cancerous tissue [1,3]. Radiation is commonly combined with trastuzumab, a HER2-targeted monoclonal antibody that binds to the IV domain of HER2 receptors, preventing homodimerization of HER2 receptors and inhibiting the MAPK and PI3K/Akt pathways [4,5,6]. In the preclinical setting, HER2 expression has been found to have a negative effect on radiation response, highlighting the need to optimize anti-HER2 targeted therapies with clinically relevant radiation therapy [7,8].

Potent longitudinal radiosensitizers must improve ROS production and accommodate oxygenation-induced changes in DNA damage response. Jongen et al. and Begg et al. demonstrate that hypoxic tumor regions are often associated with decreased expression of DNA repair proteins and decreased response to chemotherapy and radiation [9,10]. Similarly, Liu et al. notes that hypoxic conditions trigger E2F4 binding, which mechanistically decreases Rad51 non-homologous end joining, suggesting that an increase in tumor oxygenation may improve tumor DNA damage response [11]. An effective radiosensitizer must induce ROS production, while accounting for the increased DNA damage, highlighting the need for a multi-faceted cellular and molecular-based radiosensitizer.

Typical approaches for determining the efficacy of radiation therapy use clonogenic assays or flow cytometry against γ-H2AX [12,13,14]. These studies, however, lack the ability to noninvasively monitor in vivo changes in tumor oxygenation and do not account for longitudinal changes in treatment synergy. The positron emission tomography (PET) radiotracer [^18^F]-fluoromisonidazole ([^18^F]-FMISO) is used to assess severe hypoxia in vivo [15]. When injected, [^18^F]-FMISO will accumulate and irreversibly bind to thiol-rich regions of hypoxic cells which lack the oxygenation levels to oxidize the [^18^F]-FMISO [15]. Clinically, [^18^F]-FMISO PET imaging has been used as a diagnostic tool for characterizing tumor phenotype and has been shown to correlate with tumor aggression and tumor grade [16,17]. Previous preclinical studies have observed that [^18^F]-FMISO PET imaging can longitudinally quantify significant reductions in tumor hypoxia and predict response to trastuzumab in mouse models of HER2+ breast cancer prior to significant changes in tumor volume [18,19]. As tumor hypoxia prevents the formation of ROS, the ability to noninvasively monitor changes in tumor oxygenation provides an avenue to guide radiation therapy to increase treatment efficacy and reduce patient toxicity.

The overall goal of this study is to investigate the ability of HER2-targeted therapies to alter oxygenation levels and DNA damage response and enhance the effectiveness of fractionated radiation utilizing noninvasive [^18^F]-fluoromisonidazole PET imaging to longitudinally monitor tumor response. We hypothesize that [^18^F]-FMISO PET imaging can identify trastuzumab-induced increases in tumor oxygenation, which can be exploited to improve the therapeutic efficacy of fractionated radiation. Indeed, this study incorporates: (1) in vitro cell death and double-strand break (DSB) quantification assays used to evaluate the combination of trastuzumab and radiation therapy, (2) PET imaging of hypoxia changes during combination HER2-targeted and radiation therapy, and (3) in vivo monitoring of longitudinal tumor changes in targeted therapy plus fractionated radiation in xenograft cell line and patient-derived xenograft (PDX) models. PDX models are known to better recapitulate clinical characteristics of human tumors and will provide validation for future clinical translation and results from this study can be readily translated into other forms of HER2-overexpressing cancers.

## 2. Results

### 2.1. Trastuzumab Prior to Fractionated Radiation Enhances In Vitro Cytotoxicity Compared to Single Agent Therapy

Figure 1 reveals in vitro cell viability in BT474, SKBR3, MDA-MB-361, and MDA-MB-453 cell lines when treated with fractionated radiation. BT474 cells exhibited 50.3 ± 9% cell death on day 7, while BT474 cells treated with trastuzumab prior to fractionated radiation exhibited 76 ± 5.3% cell death (*p* = 0.01). When treated with single-dose radiation, BT474 cells exhibited 76.3 ± 6.1% cell death on day 7, while BT474 cells treated with trastuzumab prior to single-dose radiation exhibited 69.3 ± 2.1% cell death (*p* = 0.13) (Figure 1B).

This trend was replicated in MDA-MB-361, SKBR3, and MDA-MB-453 cell lines. When treated with fractionated radiation, MDA-MB-361 cells exhibited 53.6 ± 9% cell death, while MDA-MB-361 cells treated with trastuzumab prior to fractionated radiation exhibited 72.5 ± 2.5% cell death (*p* = 0.03). When treated with single-dose radiation, MDA-MB-361 exhibited 76.8 ± 2.5% cell death, while MDA-MB-361 cells treated with trastuzumab prior to single-dose radiation exhibited 78.3 ± 4.3% cell death (*p* = 0.65) (Figure 1C). When treated with fractionated radiation, SKBR3 cells exhibited 30.3 ± 2.3% cell death, while SKBR3 cells treated with trastuzumab prior to fractionated radiation exhibited 62.3 ± 1.5% cell death (*p* = 0.01). When treated with single-dose radiation, SKBR3 cells exhibited 54 ± 2.6% cell death, while SKBR3 cells treated with trastuzumab prior to single-dose radiation exhibited 62.3 ± 2.1% cell death (*p* = 0.01) (Figure 1D). When treated with fractionated radiation, MDA-MB-453 cells exhibited 19.7 ± 0.6% cell death, while MDA-MB-453 cells treated with trastuzumab prior to fractionated radiation exhibited 61 ± 6.1% cell death (*p* = 0.01). When treated with single-dose radiation, MDA-MB-453 cells exhibited 59 ± 2.6% cell death, while MDA-MB-453 cells treated with trastuzumab prior to single-dose radiation exhibited 57.7 ± 0.6% cell death (*p* = 0.44) (Figure 1E).

Cell death was observed in all groups treated with single-dose radiation or fractionated-dose radiation (Figure 1); however, treatment synergy was only observed in trastuzumab prior to fractionated-dose radiation in all experimental conditions.

### 2.2. Phosphorylated γ-H2AX Analysis Reveals Significantly Increased DNA Damage in Trastuzumab-Pretreated HER2+ Cancer Cells Treated with Fractionated Radiation In Vitro

In response to fractionated radiation, 9.2 ± 7.2% of BT474 cells experienced DNA damage, while 40.8 ± 11% of BT474 cells experienced DNA damage when treated with trastuzumab 24 h prior to 2 Gy × 3 fractionated radiation (*p* = 0.01) (Figure 2B,D). This trend was replicated in MDA-MB-361 cells, where, in response to fractionated radiation, 37.3 ± 7.3% of MDA-MB-361 cells experienced DNA damage, while 62.4 ± 8.7% of MDA-MB-361 cells experienced DNA damage when treated with trastuzumab 24 h prior to 2 Gy × 3 fractionated radiation (*p* = 0.01) (Figure 2C,D).

In response to single-dose radiation, 45 ± 4.1% of BT474 cells experienced DNA damage, while 40.3 ± 4.5% of BT474 cells experienced DNA damage when treated with trastuzumab 24 h prior to 6 Gy single-dose radiation (*p* = 0.24) (Figure 2B). This trend was replicated in MDA-MB-361 cells, where in response to single-dose radiation, 66.9 ± 11.7% of MDA-MB-361 cells experienced DNA damage, while 67.8 ± 4.5% of MDA-MB-361 cells experienced DNA damage when treated with trastuzumab 24 h prior to 6 Gy single-dose radiation (*p* = 0.90) (Figure 2C). A Bliss test of independence revealed treatment synergy between trastuzumab and fractionated radiation against DNA DSB in HER2+ breast cancer cells.

### 2.3. Changes in Tumor Hypoxia and Volume in Response to Treatment Synergy of Trastuzumab and Fractionated Radiation Monitored with [^18^F]-FMISO PET Imaging

In MDA-MB-361 tumors, tumors treated with trastuzumab and fractionated radiation experienced a 0.20 ± 0.08 SUV_mean_ (*p* = 0.05, compared to control), while fractionated radiation-only tumors experienced a 0.31 ± 0.06 SUV_mean_ on day 7 (*p* = 0.78, compared to control) (Figure 3A,B). It was indicated by [^18^F]-FMISO PET imaging that BT474 tumors treated with trastuzumab and fractionated radiation therapy had a 0.14 ± 0.04 SUV_mean_ (*p* = 0.04, compared to control), while tumors treated with fractionated radiation revealed a 0.20 ± 0.09 SUV_mean_ (*p* = 0.54, compared to control) (Appendix A). In BCM 3472 tumors, tumors treated with trastuzumab and fractionated radiation had a 0.24 ± 0.05 SUV_mean_ (*p* = 0.16, compared to control), while fractionated radiation-only tumors experienced a 0.23 ± 0.03 SUV_mean_ (*p* = 0.25, compared to control) (Figure 3D,E). From tumors treated with fractionated radiation, one MDA-MB-361 replicate was identified as an outlier on day 7 and was excluded from the analysis. No replicates from the BT474 and BCM 3472 PDX model were excluded from the analysis.

In Figure 4A, MDA-MB-361 tumors treated with trastuzumab and fractionated radiation also exhibited a significant reduction in hypoxic fraction compared to control tumors on day 7 (76.3 ± 16%, *p* = 0.05) (Figure 4A). Moreover, MDA-MB-361 control tumors exhibited a 93.7 ± 6.6% hypoxic fraction on day 7, while tumors treated with trastuzumab showed a significant reduction in hypoxic fraction (67.1 ± 17%, *p* = 0.01). In comparison to MDA-MB-361 control tumors, MDA-MB-361 tumors treated with radiation therapy exhibited an 87.7 ± 10% hypoxic fraction on day 7 (*p* = 0.31) (Figure 4B).

In BT474 models, BT474 tumors treated with trastuzumab and fractionated radiation had significantly lower hypoxic fraction on day 7 compared to controls (34.0 ± 26.8% *p* = 0.04) (Appendix A). Moreover, control tumors had 72.1 ± 26.3% hypoxic fraction on day 7, while tumors treated with trastuzumab had a reduced hypoxic fraction, trending towards significance, of 30.6 ± 34.6% on day 7 (*p* = 0.06). In comparison with BT474 control tumors, BT474 tumors treated with radiation therapy exhibited a 64.3 ± 37.2% hypoxic fraction on day 7 (*p* = 0.70) (Appendix A). In BCM 3472 tumor models, BCM 3472 PDX tumors treated with trastuzumab and fractionated radiation had a 78.2 ± 7.9% hypoxic fraction on day 7 (*p* = 0.01, compared to control) (Figure 4F). Control tumors had a 38.9 ± 21.8% hypoxic fraction on day 7, while tumors treated with trastuzumab had 64.6 ± 37.9% on day 7 (*p* = 0.28). In comparison with BCM 3472 control tumors, BCM 3472 tumors treated with radiation therapy exhibited 74.6 ± 7.5% hypoxic fraction on day 7 (*p* = 0.02) (Figure 4D).

In MDA-MB-361 models, tumors treated with trastuzumab and fractionated radiation experienced a 23.9 ± 14.3% decrease in tumor volume (*p* = 0.003, compared to control), while tumors treated with fractionated radiation experienced a 10.4 ± 2.8% decrease in tumor volume from day 0 to day 7 (*p* = 0.03, compared to control) (Figure 3C). In BT474 tumor models, tumors treated with trastuzumab and fractionated radiation experienced a 47.1 ± 15.7% decrease in tumor volume (*p* = 0.0001, compared to control), while tumors treated with fractionated radiation which showed a 19.8% decrease in tumor volume from day 0 to day 7 (*p* = 0.17, compared to control) (Appendix A). In BCM 3472 tumor models, tumors treated with trastuzumab and fractionated radiation experienced a 46% ± 15.7% decrease in tumor volume (*p* = 0.03, compared to control), while tumors treated with fractionated radiation experienced a 17.5% ± 32% decrease in tumor volume from day 0 to day 7 (*p* = 0.82, compared to control) (Figure 3F).

### 2.4. Trastuzumab Synergistically Improves the Efficacy of Fractionated Radiation over the Longitudinal Observation Period

Figure 5A,B detail longitudinal changes in tumor viability after treatment with trastuzumab and fractionated radiation. Tumor volumes were normalized to the start of treatment (day 45 for MDA-MB-361; day 21 for BCM 3472). MDA-MB-361 tumors treated with a control had a 147.8 ± 62.9% change in tumor growth. MDA-MB-361 tumors treated with single-agent trastuzumab or fractionated radiation had a percent change of 101.6 ± 94.4% and 140.2 ± 58.6%, respectively (*p* = 0.46 and *p* = 0.57, compared to controls) (Figure 5A). MDA-MB-361 tumors treated with trastuzumab prior to fractionated radiation had a normalized percent change of −62 ± 62.9% (*p* = 0.03, compared to radiation-treated tumors). Analysis of tumor volume reveals similar trends in tumor burden reduction in those treated with trastuzumab prior to fractionated radiation (Appendix A).

To expand the clinical relevance of our experiment, tumor samples from a trastuzumab-responsive patient (BCM3472) were obtained and serially passaged in mice prior to this experiment. BCM3472 is a G3 poorly differentiated pathology graded tumor that was biopsied from an infiltrating ductal carcinoma. The patient was 47 years old at the time of excision and the tumor is ER–, PR–, and HER2+ [20]. Four weeks after treatment, BCM 3472 control tumors had an end volume of 839 ± 281 mm^3^. BCM 3472 treated with a control had a percent change of 263.7 ± 114.1% in tumor growth. BCM 3472 tumors treated with single-agent trastuzumab or fractionated radiation had a percent change of 327.8 ± 206.6% and 7.7 ± 27.2%, respectively (*p* = 0.65 and *p* = 0.06, compared to controls) (Figure 5B). BCM 3472 tumors treated with trastuzumab prior to fractionated radiation had a normalized percent change of −68.2 ± 2.3% (*p* = 0.03, compared to radiation-treated tumors). Analysis of tumor volume reveals similar trends in tumor burden reduction in replicates treated with trastuzumab prior to fractionated radiation (Appendix A). A Bliss test of independence confirmed the treatment synergy of trastuzumab and fractionated radiation, starting 14 days after the start of trastuzumab therapy.

### 2.5. Analyte Analysis Reveals Decreased Expression of IL-2 and IFN-Gamma in Cytotoxic Combination-Treated Tumors

Figure 5C details mechanisms of sustained synergy through immune cell activation in response to trastuzumab and fractionated radiation through IL-2 expression. Control tumors had an average IL-2 expression of 94.9 ± 56.7 pg/mL. Tumors treated with single-agent trastuzumab therapy had an average IL-2 expression of 115.9 ± 83.8 pg/mL (*p* = 0.55, compared to control) and tumors treated with fractionated radiation had an average IL-2 expression of 89.5 ± 78.4 pg/mL (*p* = 0.88, compared to control). Tumors treated with trastuzumab and fractionated radiation had an average IL-2 expression of 54.9 ± 28.2 pg/mL (*p* = 0.04, compared to control).

Figure 5D details mechanisms of sustained synergy through immune cell activation in response to trastuzumab and fractionated radiation through IFN-gamma expression. Control tumors had an average IFN-gamma expression of 244.1 ± 71.9 pg/mL. Tumors treated with single-agent trastuzumab therapy had an average IFN-gamma expression of 322.4 ± 133.8 pg/mL (*p* = 0.17, compared to control) and tumors treated with fractionated radiation had an average IFN-gamma expression of 261.9 ± 156.9 pg/mL (*p* = 0.73, compared to control). Tumors treated with trastuzumab and fractionated radiation had an average IFN-gamma expression of 170.9 ± 84.8 pg/mL (*p* = 0.03, compared to control).

### 2.6. Nanostring Analysis Reveals Increased Expression of Vascular and Immune Modulation in Combination-Treated Tumors

Figure 5E demonstrates mechanisms of hypoxia and immune modulation in response to trastuzumab and fractionated radiation as extracted from a Nanostring transcriptomic analysis. Tumors treated with single-agent trastuzumab or single-agent fractionated radiation had a 4.0 linear fold increase and 4.3 linear fold increase in THBS-4 expression relative to control tumors, respectively (*p* < 0.01 and *p* = 0.11); however, tumors treated with trastuzumab prior to fractionated radiation had a 8.0 linear fold change in THBS-4 expression relative to control tumors (*p* < 0.01). Nanostring results were validated by THBS-4 IHC (Figure 5F–G). Tumors treated with single-agent trastuzumab or single-agent fractionated radiation exhibited 15.1% THBS-4 staining and 16.4% THBS-4 staining, respectively (*p* = 0.0006 and *p* = 0.04, relative to control tumors); however, tumors treated with trastuzumab prior to fractionated radiation had 22.4% THBS-4 staining, relative to control tumors (*p* = 0.01, relative to control tumors).

### 2.7. Immunohistochemical Analysis Reveals Decreased Oxygenation in Trastuzumab Monotherapy and Cytotoxic Combination-Treated Tumors

Figure 6A shows qualitative and quantitative immunohistochemical analysis after treatment with trastuzumab and fractionated radiation. In MDA-MB-361 tumor models, MDA-MB-361 control tumors exhibited 27.6 ± 7.4% hypoxia, while tumors treated with the combination of trastuzumab and fractionated radiation exhibited 12.3 ± 8.6% hypoxia per viable tissue (*p* = 0.05) (Figure 6B). MDA-MB-361 tumors treated with trastuzumab showed non-significant decreases in hypoxia per viable tissue relative to control (*p* = 0.12). In BT474 tumor models, BT474 control tumors exhibited 22.2 ± 9.9% hypoxia, while tumors treated with trastuzumab exhibited 6.4 ± 3.9% hypoxia per viable tissue (*p* = 0.01) (Figure 6C). Evaluation of tissue necrosis showed that no significant changes in tumor necrosis were observed in tumors treated with trastuzumab, fractionated radiation, or trastuzumab prior to fractionated radiation (Figure 6D,E).

## 3. Discussion

In this study, we evaluated the efficacy of anti-HER2 therapy to sensitize HER2+ breast cancer to fractionated radiation at both the cellular and molecular level with in vitro and in vivo cell-line models and PDX models of HER2+ breast cancer. Cellular response to fractionated radiation results in increased cell death and increased DNA DSB when cells are pretreated with a low, non-therapeutic dose of trastuzumab in vitro. Furthermore, when evaluating the synergy of these therapies in vivo there was sustained and long-term anti-tumor response in both cell-line xenograft and PDX models that continued well after treatment was halted. Hypoxia PET imaging was able to quantitatively assess decreases in hypoxia in the tumors treated with trastuzumab, as a monotherapy and when given first in cytotoxic combination therapy, as confirmed by pimonidazole IHC staining. It is noted that the role of hypoxia in ROS generation is contradictory; however, our results find reoxygenation of the tumor microenvironment in combination with radiation therapy induces a state of longitudinal sustained response. These results may inform and improve the prognosis of patients with hypoxic radioresistance, where cancer cells gain resistance to ROS due to lack of available oxygenation [21]. To our knowledge, this is the first study that has evaluated the synergy of treatment efficacy of HER2-targeted therapy and radiation treatment in vivo, quantified changes in the sequencing of the combination treatment of trastuzumab plus fractionated radiation in vitro, and utilized molecular imaging to evaluate longitudinal alterations in the tumor microenvironment.

Molecular imaging has been utilized to evaluate changes in HER2+ breast cancer to predict eventual response [18,19,22,23,24,25,26,27,28]. Whisenant et al. used [^18^F]-FLT-PET imaging to monitor trastuzumab therapeutic response in tumor proliferation in trastuzumab-sensitive and -resistant models and found that [^18^F]-FLT-PET imaging could be used to inform longitudinal trastuzumab sensitivity as early as 24 h after the first therapeutic dose [25]. Similarly, a HER2+ trastuzumab-resistant model would provide additional biological understanding of the mechanisms of synergy between trastuzumab and radiation therapy. Future expansion of this may help to identify HER2-specific induced changes in the tumor microenvironment.

In previous studies, higher doses of trastuzumab (such as 10 mg/kg) were used to mimic clinical doses and were observed to improve tumor oxygenation [18,23,25,26,29]; however, other studies have reported using up to 35 mg/kg to show sustained response to trastuzumab [26,30]. We previously observed that treating an in vivo model of HER2+ breast cancer with 10 mg/kg trastuzumab prior to doxorubicin significantly improved tumor vascular delivery and sensitized the tumor to chemotherapy [23], therefore significantly altering sustained tumor burden long after therapy was stopped. In this study, we observed that a subtherapeutic dose (4 mg/kg) could be used to significantly decrease tumor hypoxia and modulate the tumor microenvironment in xenograft models of HER2+ breast cancer on day 7 and synergistically improves radiation therapy 14 days after the start of therapy (*p* < 0.05). In PDX models of HER2+ breast cancer, while a subtherapeutic dose of trastuzumab did not show significant decreases in tumor hypoxia, treatment synergy with radiation therapy was still observed. FMISO-PET results are supported by pimonidazole IHC staining for hypoxia, which has been shown to be highly specific for low levels of oxygenation. To further evaluate the entire pathway of hypoxia, hypoxia-induced HIF1-alpha Western blot analysis would provide additional upstream details. Furthermore, while the PDX patient of origin is trastuzumab-responsive, the PDX model did not demonstrate reduced tumor volume when treated with low doses of trastuzumab alone [31]. While not directly studied, the synergy of these therapies may be irrespective of ER and PIK3CA mutations, as the cell lines investigated had a wide range of expression characteristics and the overall study would be bolstered by additional assessment of cell-line genomic characteristics and its relationship to overall treatment synergy. This data provides additional information on pathways that may not have a direct involvement in the synergy of trastuzumab and radiation. Utilizing low trastuzumab doses, in addition to increasing the efficacy of radiation therapy, has the potential to significantly impact tumor response, while reducing risk for trastuzumab-related cardiotoxicity, thus raising patient quality of life [32,33].

Although studies evaluating the effects of trastuzumab and radiation therapy are limited, our results are corroborated by studies evaluating in vitro cell viability in response to trastuzumab and radiation. Liang et al. used an ELISA assay against apoptosis and found increased cell death in HER2+ breast cancer when treated with trastuzumab and single-dose radiation [14]. Furthermore, Liang et al. transfected the MCF-7 cell line with a HER2 cDNA plasmid and found that trastuzumab-induced downregulation of PI3-K pathways is involved in radiosensitization. As hypoxia has been reported to inhibit DNA damage repair, synergy between trastuzumab and fractionated radiation therapy is supported by molecular imaging, which shows that trastuzumab reduces tumor hypoxia, and cellular assays, which show that trastuzumab prior to fractionated radiation increases DNA DSB. In our study, it was reported that trastuzumab increased DNA DSB in response to fractionated radiation; however, no increase in DNA damage was observed in single-dose radiation groups. It is hypothesized that, at 6 Gy, cells sustain larger amounts of DNA damage that result in increased cell death. Future experiments should confirm such results by studying the effect of trastuzumab on radiation-induced mitotic arrest through clonogenic assays. Results from our study indicate that trastuzumab radiosensitization may be multifaceted, including both cellular and tumor microenvironment alterations, which is emphasized in this study showing both in vitro and in vivo alterations.

To study immune-based mechanisms of long-term sustained synergy in in vivo models of breast cancer, we conducted immunophenotyping of IL-2, IFN-gamma, and THBS-4 in treated tumors with cytokine and Nanostring transcriptomic analysis. Since the human cell line and PDX models were conducted in an immunocompromised model, we focused on M1 macrophage and natural killer cell activity. Interestingly, we observed significant decreases in IL-2 and IFN-gamma expression in cytotoxic combination-treated tumors, while long-term synergy was observed. High doses of IL-2 have been reported to limit natural killer cell activation through selective expansion of regulatory T-cells, suggesting increased natural killer cell efficacy with decreased doses of IL-2 [34]. Similarly, IFN-gamma has been observed to exhibit contradictory tumor-suppressive and tumor-promoting properties. Benci et al. observed that increased IFN-gamma signaling is associated with resistance to checkpoint inhibitor immunotherapy through changes in STAT-1 epigenetic changes that promote immunoevasion [35]. Our results suggest that changes in IL-2, IFN-gamma, and THBS-4 expression may sensitize the tumor to long term sustained treatment response by triggering a state of enhanced immune activation. The overall impact of anti-HER2 induced radiosensitization on the greater immune system should be further studied in a humanized model of HER2+ breast cancer and probing of CD4+ and CD8+ T-cell trafficking. Additional investigations into the connection between T-cells, B-cells, and radiotherapeutic response can also be conducted in the future with a syngeneic model; however, there would be additional challenges procuring a mouse version of trastuzumab, which is not commercially available. Moreover, in a syngeneic tumor model, the role of B-cells and T-cells in radiosensitization can be investigated by selective depletion of adaptive and innate immune systems [36].

This study used in vivo [^18^F]-FMISO-PET imaging to determine whether changes in tumor hypoxia could be exploited with clinically relevant fractionated radiation. These exciting results provide support for future studies that may examine alterations in the dosing of both radiation and trastuzumab therapy, such as evaluating if a low dose of trastuzumab prior to fractionated radiation has equivalent tumor kill to a single higher dose of radiation or if increased doses of trastuzumab will increase synergy further. Future studies using PDX models could investigate whether a clinically relevant dose of trastuzumab (10 mg/kg) could be used to increase oxygenation in this more clinically representative model. Since we saw synergy in a model that did not respond to sub-therapeutic dosing of trastuzumab, this leads to the interesting question of whether this approach would work for trastuzumab-resistant (acquired or de novo) tumors. While it is not standard of care to give trastuzumab in combination with radiation therapy in locally advanced HER2+ breast cancer, this manuscript outlines the important relationship between HER2 and radiation sensitivity, which can be used to benefit patients with chest-wall or locoregional recurrences, oligometastatic disease and brain metastases.

## 4. Materials and Methods

### 4.1. HER2+ Cell Culture

The cell lines used in this study include BT474 (estrogen receptor (ER)+, HER2+, Phosphatidylinositol-4,5-Bisphosphate 3-Kinase Catalytic Subunit Alpha (PIK3CA) mutant), SKBR3 (ER-, HER2+, PIK3CA WT), MDA-MB-361 (ER+, HER2+, PIK3CA mutant), and MDA-MB-453 (ER-, HER2+, PIK3CA mutant) [37]. BT474 breast cancer cells were grown in improved minimal essential media (IMEM, Invitrogen, Carlsbad, CA, USA) supplemented with 10% FBS and 1% insulin. SKBR3 breast cancer cells were grown in McCoy’s 5A medium (ATCC, Manassas, VA, USA) supplemented with 10% FBS, 2 mM L-glutamine, and 1 mM sodium pyruvate. MDA-MB-361 and MDA-MB-453 breast cancer cells were grown in Dulbecco’s Minimal Essential Medium (Thermo Fisher Scientific Inc., Waltham, MA, USA) supplemented with 20% FBS, 2 mM L-glutamine, and 1 mM sodium pyruvate.

### 4.2. Tumor Model

All animal procedures were approved by our institution’s animal care and use committee under IACUC APN#: 21507. All animals used in this experiment were housed and monitored in accordance with the University of Alabama at Birmingham’s Animal Resources Program. For cell line-based tumor models, six-week-old female athymic nude mice were obtained from Charles Rivers Labs (catalog number: 490) and were subcutaneously implanted with 0.72 mg 17β-estradiol pellets. Approximately 24 h later, 10^7^ BT474 cells (N = 24), in serum-free IMEM and 30% Matrigel, or 10^7^ MDA-MB-361 (N = 19), in serum-free DMEM and 50% Matrigel, were subcutaneously injected into the right shoulder of the mouse.

For PDX models, six-week-old female NOD.Cg-Prkdc^scid^Il2rg^tm1Wjl^ISzJ (NSG) mice were obtained from Jackson Labs (catalog number: 005557) and subcutaneously implanted with 0.72 mg 17β-estradiol pellets under 2% isofluorane anesthesia. Approximately 24 h later, 100–150 mm^3^ tumor pieces of BCM 3472 (HER2+) were surgically engrafted into the cleared third mammary fat pad with Matrigel supplement under 2% isofluorane anesthesia [31].

Tumors were monitored on a weekly basis and enrolled into the study once they reached approximately 225 mm^3^. All tumors were randomized and treated with either (1) saline, (2) trastuzumab (4 mg/kg), (3) fractionated radiation (2 Gy × 3), or (4) a combination of trastuzumab (4 mg/kg) and fractionated radiation (2 Gy × 3). Trastuzumab was administered via intraperitoneal injection on days 0 and 3. Fractionated radiation was administered via a X-RAD 320 irradiator (Precision X-ray, North Bradford, CT, USA) in 2 Gy fractions on days 1, 2, and 3.

#### 4.2.1. Evaluation of In Vitro Treatment Synergy between Trastuzumab and Fractionated Radiation

To examine treatment synergy between trastuzumab and radiation in cancer cell-line models, cells were treated with trastuzumab, 6 Gy single-dose radiation, 2 Gy × 3 fractionated radiation, trastuzumab → 6 Gy single-dose radiation, and trastuzumab → 2 Gy × 3 fractionated dose radiation and assayed for changes in cell viability. BT474, SKBR3, MDA-MB-453, and MDA-MB-361 cells were plated in 96-well plates at a density of 10,000 cells/well. After 24 h (day 1), cells were treated with trastuzumab (1 µg/mL) for 24 h (Figure 1A). Trastuzumab was removed and cells were irradiated with 2 Gy on days 2, 3, and 4 with an X-RAD 320 irradiator (Precision X-ray, North Bradford, CT, USA). Then, 72 h after final radiation fraction (day 7), cells were trypsinized and assessed for changes in cell viability with a trypan blue live cell-exclusion assay. Percent cell death in each experimental condition was derived by normalizing the number of living cells in treated groups by the number of living cells in the control group. Each group has 4 replicates.

#### 4.2.2. Evaluating In Vitro Treatment Synergy through Flow Cytometry against DNA Repair

To determine how trastuzumab increases the efficacy of fractionated radiation therapy in vitro, cells were treated with trastuzumab, 6 Gy single-dose radiation, 2 Gy × 3 fractionated dose radiation, trastuzumab → 6 Gy single-dose radiation, and trastuzumab → 2 Gy × 3 fractionated-dose radiation and probed against phosphorylated γ-H2AX with flow cytometry. BT474 and MDA-MB-361 cells were plated in a 6-well plate at a density of 160,000 cells/well at baseline (day 0). After 24 h (day 1), cells were treated with trastuzumab (1 µg/mL). On day 2, trastuzumab was removed and cells were irradiated with 2 Gy on days 2, 3, and 4 using the same parameters as indicated above (Figure 2A). Then, 30 min after final radiation fraction, cells were trypsinized and fixed with 1% formalin for 10 min. Cells were washed with PBS and incubated in cold 70% ethanol for 20 min. Cells were washed with PBS and stained with rabbit anti-human phosphorylated γ-H2AX Ser139 (Cell Signaling Technology, Beverly, MA, USA) for 2 h. Cells were washed with PBS and stained with Alexa Fluor 488 #A32723 (Thermo Fisher Scientific, Inc., Waltham, MA, USA) goat anti-rabbit for one hour in the dark. Cells were then washed and resuspended in a solution of 0.2% BSA in PBS and analyzed with an Attune N × T flow cytometer. Each treatment group had 3 wells of replicates.

#### 4.2.3. Non-Invasive [^18^F]-FMISO PET Imaging Identifies Increases in Tumor Oxygenation That Can Be Exploited with Fractionated Radiation Therapy In Vivo

To determine whether trastuzumab-induced tumor reoxygenation synergistically increases the efficacy of fractionated radiation in vivo, BT474, MDA-MB-361, and BCM 3472 tumors were treated with 4 mg/kg trastuzumab, 2 Gy × 3 fractionated radiation, or a combination of 4 mg/kg trastuzumab prior to 2 Gy × 3 fractionated radiation and were imaged with static [^18^F]-FMISO PET. The [^18^F]-FMISO was synthesized by the University of Alabama at Birmingham cyclotron facility on a GE FASTlab2 or a Synthra RNplus synthesizer according to literature procedures [38,39]. Mice with BT474, MDA-MB-361, and BCM 3472 tumors (N = ~20 per model, N = 4–7 per condition) were imaged with [^18^F]-FMISO PET on days 0, 3, and 7. At each imaging time point, the mice were administered approximately 150 µCi (154.6 ± 26.3 µCi) of [^18^F]-FMISO via retro-orbital injection and were imaged with a F-18 static 20 min PET scan and 80 kVp Bin 2 CT with a micro-PET/CT (Sofie Biosciences, Somerset, NJ, USA) 80 min later. Images were processed and quantified through VivoQuant (InviCRO, Boston, MA, USA). PET regions of interest (ROI) were drawn on CT images and overlaid onto PET images to calculate mean standard uptake value (SUV_mean_) for each tumor. To determine the fraction of hypoxic and normoxic cells, voxel-wise SUV was quantified through VivoQuant’s 3D ROI histogram analysis. Hypoxic threshold was determined per model with the equation: muscle SUV_mean_ + 1 standard deviation [40]. Voxels above this threshold were classified as “hypoxic” and voxels below this threshold were classified as “normoxic”. BT474 and MDA-MB-361 tumors were collected following the final [^18^F]-FMISO PET imaging timepoint and sectioned for Bioplex cytokine detection assay and immunohistochemistry.

#### 4.2.4. Longitudinal Monitoring of Changes in Tumor Viability in Response to Trastuzumab and Fractionated Radiation Therapy In Vivo

To determine whether trastuzumab synergistically increases the long-term efficacy of fractionated radiation therapy, MDA-MB-361 and BCM 3472 tumors were treated with trastuzumab, fractionated radiation, or a combination of trastuzumab prior to fractionated radiation. MDA-MB-361 and BCM 3472 tumors were measured for four weeks (experimental time point). Once mice reached the experimental endpoint, the mice were euthanized, and tumors were excised.

### 4.3. Cytokine Detection Assay

BT474 and MDA-MB-361 tumors were collected following the final [^18^F]-FMISO PET imaging timepoint. Tumors were flash-frozen, mechanically dissociated, and lysed with Cell Lysis Buffer (Catalog number: EPX-99999-000; Life Technologies Corporation, Grand Island, NY, USA). Protein lysate concentration was determined using the NanoDrop 2000 (Thermo Fisher Scientific, Waltham, MA, USA). The cytokine detection assay was performed on the Bioplex 2000 (BioRad Laboratories, Hercules, CA, USA). Samples were probed against IL-2 (Catalog number: 171G5003M. BioRad Laboratories, Hercules, CA, USA) and IFN-gamma (Catalog number: 171G5017M. BioRad Laboratories, Hercules, CA, USA).

### 4.4. Nanostring RNA Analysis

Following [^18^F]-FMISO-PET imaging, BT474 tumors were analyzed using Nanostring RNA analysis. RNA was isolated from FFPE histology slides using the PureLink FFPE RNA isolation kit (Catalog number: K156002. Thermo Fisher Scientific, Waltham, MA, USA) and was quantified with the Nanostring Human Pancancer Pathways Panel and nSolver 4.0 with Advanced Analysis 2.0 software.

### 4.5. Immunohistochemistry

BT474 and MDA-MB-361 tumors were collected following [^18^F]-FMISO PET imaging on day 7. BCM 3472 and an additional cohort of MDA-MB-361 tumors were collected 30 days after initial treatment. Mice were injected with 1.5 mg pimonidazole (Hypoxyprobe, Burlington, MA, USA) one hour prior to tumor excision. Tumors were cut at the largest cross-section corresponding to the in vivo imaging plane, fixed in 10% formalin, and transferred to 70% ethanol for immunohistochemistry. Each tumor was stained with hematoxylin and eosin (H&E), 1:300 anti-pimonidazole, and 1:100 anti-THBS4 (Catalog number: MAB2390; R&D Biosystems, Minneapolis, MN, USA). Hypoxia was determined according to the percentage of the overall viable tumor that is stained with pimonidazole. Anti-THBS4 was quantified by randomly selecting five 513 × 382 micron regions per tumor and percent of signal intensity was quantified with Matlab custom scripts.

### 4.6. Statistical Analysis

In vitro cell death in treated groups was normalized to control groups. Groups were summarized by average cell death, average percentage of cells with DNA DSB, and standard error of the mean (SEM). A Wilcoxon rank sum test was used to assess group differences. The Grubbs outlier test eliminated data points that were statistical outliers. All data and figures were analyzed using GraphPad Prism 7 (La Jolla, CA, USA).

## 5. Conclusions

This study reveals that a subtherapeutic dose of anti-HER2 trastuzumab therapy can synergize with clinically relevant fractionated radiation therapy to reduce tumor burden and increase cytotoxic efficacy. Both cellular (increased DSB) and molecular changes (reduced tumor hypoxia) result in a more sensitized tumor state as shown by in vitro and in vivo studies. Such results highlight the potential for this therapeutic combination in patients regardless of initial HER2 tumor grading or baseline response to anti-HER2 therapy. Results from this study can provide a framework for other types of cancer that overexpress HER2 or undergo first-line radiation therapy (i.e., gastric and lung cancer) to increase therapeutic efficacy, reduce tumor burden, and provide clinically relevant noninvasive monitoring approaches of the tumor microenvironment to help guide clinical decision making in cancer.

## Figures and Tables

**Figure 1 cancers-14-01015-f001:**
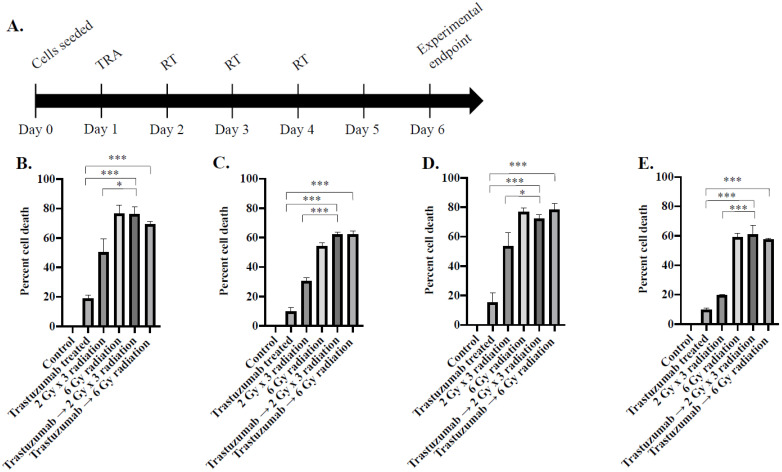
Timeline (**A**) and longitudinal assessment of cell viability in (**B**) BT474, (**C**) SKBR3, (**D**) MDA-MB-361, and (**E**) MDA-MB-453 cell lines in response to 1 ug/mL trastuzumab and radiation in vitro. Bliss test of synergy reveals a synergistic relationship between trastuzumab (TRA) and fractionated radiation (RT) in four HER2+ breast cancer cell lines. * denotes *p* < 0.05. *** denotes *p* < 0.01.

**Figure 2 cancers-14-01015-f002:**
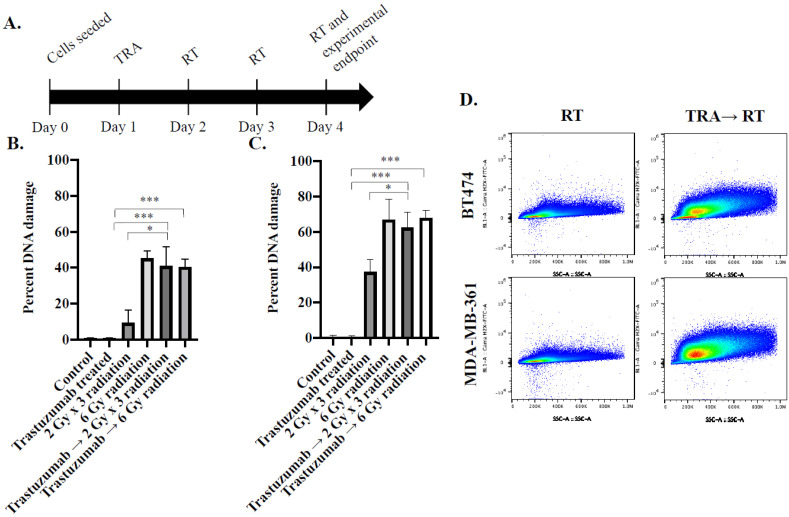
Timeline (**A**) and flow cytometry analysis of trastuzumab and radiation synergy through analysis of DNA DSB in BT474 (**B**) and MDA-MB-361 (**C**). Representative panels of BT474 and MDA-MB-361 (**D**) show the changes in DNA damage when cells are pre-treated with trastuzumab (TRA) followed by radiation therapy (RT). * denotes *p* < 0.05. *** denotes *p* < 0.01.

**Figure 3 cancers-14-01015-f003:**
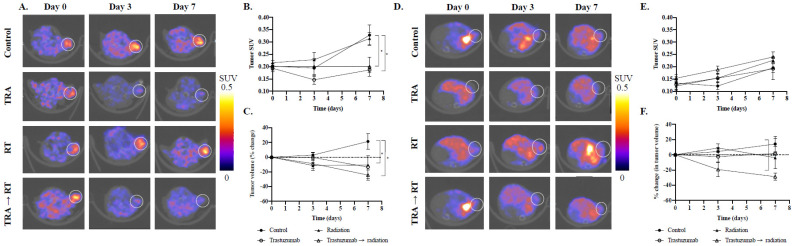
Quantitative [^18^F]-FMISO PET imaging of MDA-MB-361 and BCM3472 tumors and quantitative SUV and tumor volume analysis. MDA-MB-361 tumors were treated with trastuzumab (TRA), fractionated radiation (RT), or a combination of trastuzumab and fractionated radiation and FMISO uptake was monitored (**A**). Significant decreases in tumor SUV_mean_ (**B**) were observed when tumors were treated with trastuzumab, as a monotherapy and cytotoxic combination therapy, in cell-line xenograft models, MDA-MB-361. Significant decreases in tumor volume (**C**) were observed in MDA-MB-361 tumors treated with fractionated radiation and trastuzumab and fractionated radiation-treated groups (*p* < 0.05). BCM3472 tumors were treated with trastuzumab (TRA), fractionated radiation (RT), or a combination of trastuzumab and fractionated radiation and FMISO uptake was monitored (**D**). Significant decreases in tumor SUV_mean_ (**E**) were observed when tumors were treated with trastuzumab, as a monotherapy and cytotoxic combination therapy, in cell-line xenograft model, MDA-MB-361. Significant decreases in tumor volume (**F**) were observed in BCM 3472 tumors treated with trastuzumab and fractionated radiation-treated groups (*p* < 0.05). * denotes *p* < 0.05.

**Figure 4 cancers-14-01015-f004:**
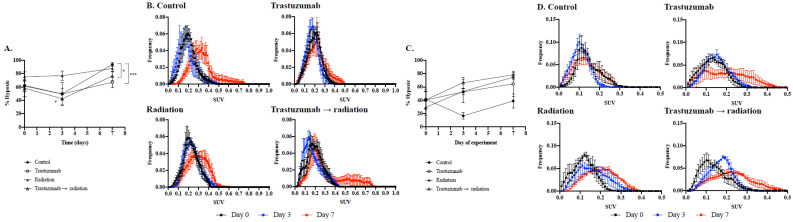
Hypoxic fraction (**A**,**C**) and histogram analysis (**B**,**D**) of MDA-MB-361 or BCM3472 tumor models. Hypoxic fraction analysis was conducted on MDA-MB-361 or BCM 3472 tumors, based on [^18^F]-FMISO SUV (**A**,**C**, respectively). Hypoxic fraction analysis revealed a significant decrease in hypoxic cells in the trastuzumab and fractionated radiation-treated group compared to the fractionated radiation-treated group (*p* = 0.01). Hypoxic fraction analysis also revealed a significant decrease in hypoxic cells in the trastuzumab-treated and trastuzumab and fractionated radiation-treated groups (*p* = 0.01 and *p* = 0.05, respectively). Error bars are representative of standard error of the mean (SEM). Histogram analysis revealed a hypoxic shift in SUV on day 7 in control tumors (**B**) and fractionated radiation-treated tumors, while SUV on day 7 slightly decreased in trastuzumab-treated and trastuzumab and fractionated radiation-treated tumors. * denotes *p* < 0.05. *** denotes *p* < 0.01.

**Figure 5 cancers-14-01015-f005:**
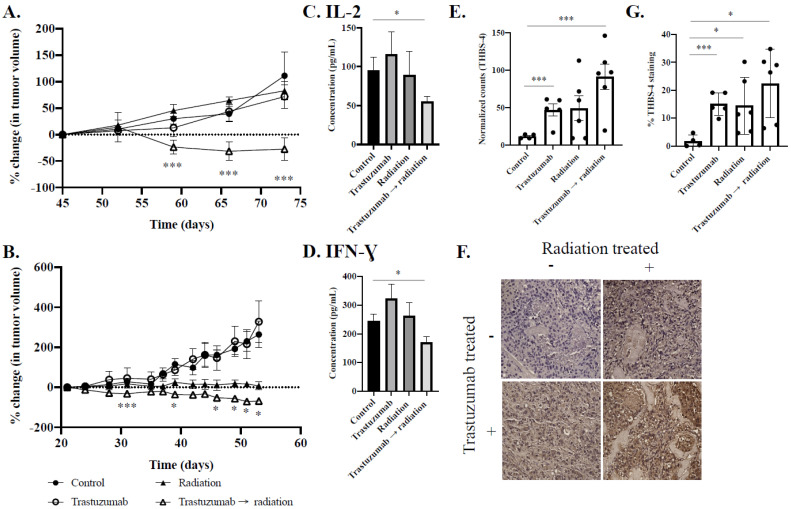
Longitudinal monitoring of normalized tumor volume in MDA-MB-361 (**A**) and PDX BCM 3472 (**B**) and mechanisms of sustained treatment synergy through IL-2 (**C**), IFN-gamma (**D**), and THBS-4 (**E**), with representative IHC (**F**) and tissue staining quantification (**G**). Significant differences in normalized tumor volume of trastuzumab- and fractionated radiation-treated tumors, compared to fractionated radiation-treated tumors, was observed beginning 14 days after the start of therapy. A Bliss test of independence confirmed treatment synergy beginning 14 days after the start of therapy. Percent change in tumor volume confirmed significant differences in tumor volume when treated with fractionated radiation vs. trastuzumab prior to fractionated radiation in MDA-MB-361 tumors (**A**) and BCM 3472 (**B**) tumors. Significant differences in IL-2 (**C**), IFN-gamma (**D**), and THBS-4 (**E**) were observed in combination trastuzumab- and fractionated radiation-treated tumors, relative to control tumors. Representative THBS-4 IHC at 20× magnification and quantitative IHC analysis reveal significant increases in THBS-4 in tumors treated with trastuzumab, fractionated radiation, and trastuzumab prior to fractionated radiation-treated tumors. * denotes *p* < 0.05. *** denotes *p* < 0.01.

**Figure 6 cancers-14-01015-f006:**
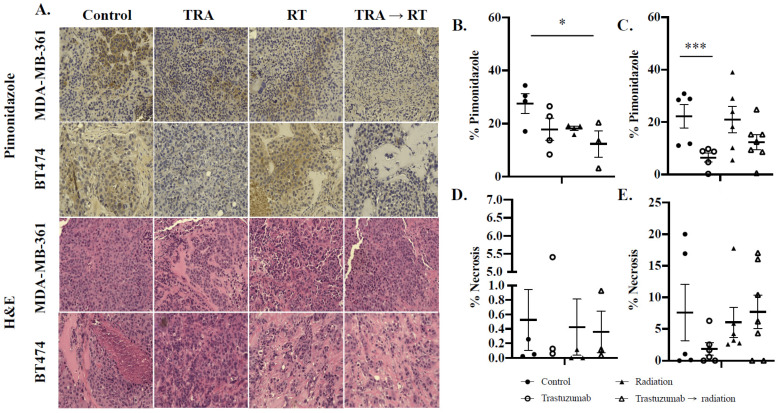
Immunohistochemical and H&E analysis of hypoxia and tumor necrosis, respectively, in MDA-MB-361 and BT474 tumor models. Representative images of immunohistochemical analysis of pimonidazole and H&E staining in MDA-MB-361 and BT474 tumors (**A**) treated with trastuzumab (TRA), radiation therapy (RT), and a combination of trastuzumab prior to radiation therapy at 20× magnification. Immunohistochemical staining of pimonidazole in MDA-MB-361 (**B**) revealed decreases in tumor hypoxia in trastuzumab monotherapy (*p* = 0.12) and trastuzumab prior to radiation therapy (*p* = 0.05). Immunohistochemical analysis of pimonidazole in BT474 (**C**) revealed decreases in tumor hypoxia in trastuzumab monotherapy (*p* < 0.01) and trastuzumab prior to radiation therapy (*p* = 0.07). H&E analysis of MDA-MB-361 (**D**) and BT474 (**E**) revealed no significant changes in necrosis in treated tumors compared to control. * denotes *p* < 0.05. *** denotes *p* < 0.01.

## Data Availability

Datasets and materials are available upon reasonable request.

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
