# Peer review of "Modulation of the Tumor Microenvironment with Trastuzumab Enables Radiosensitization in HER2+ Breast Cancer"

_cancers, 2022, doi:10.3390/cancers14041015_

Round 1
Reviewer 1 Report
Interesting concept and experimental design but there is huge scope to improve the article.
Critical comments:
- Authors use different HER2+ cell lines but failed to include trastuzumab-resistant or lapatinib-resistant cell lines
- Synergistic effect of trastuzumab and radiation was observed in all cell lines irrespective of ER or PIK3CA mutation status...any comments from authors site
- Authors may provide a table to describe the mutations profile and ER status of all cell lines.
- Before line 247, authors may provide 1 r 2 sentences about the experiments and about BCM3472
- Authors may provide the Western blots of phospho-gamma H2X
- Authors also provide the hypoxia-driven HIF1alpha status before and after treatment
- Immune part may be improved if authors include humanized PDX model or even syngeneic model
Minor comments:
- Please rectify typo errors
- It will be better to follow conclusion just after discussion
- Discussion may be little compact
Reviewer 2 Report
The authors wish to optimize the injection of trastuzumab (anti-HER2) which is proposed to improve the sensitivity of cancer cells to radiotherapy by increasing tumor oxygenation. Trastuzumab is used clinically to treat breast tumors expressing HER2. The overall objective was to use [18F] -FMISO and positron emission tomography (PET) imaging to identify increases in tumor oxygenation induced by trastuzumab which could be used to improve efficacy of fractionated radiation therapy for the treatment of breast cancer. Although interesting results were obtained, the experimental design contains the important errors.
Authors should be aware that the primary breast tumor is surgically removed prior to radiation therapy which, in combination with trastuzumab, eliminates cancer cells that have infiltrated breast tissue. The level of hypoxia is not a problem for cancer cells infiltrating the breast. In addition, trastuzumab is usually given in combination with chemotherapy followed about 5 weeks later by radiation therapy. Although it is safe to give trastuzumab at the same time as radiation therapy, these treatments are rarely given simultaneously in the clinic. This study is therefore not clinically relevant.
It is true that PDX models in immunodeficient mice can better recapitulate some clinical features of human tumors. However, the tumor response to radiation depends on the inflammatory and immunological responses induced by the treatment. An immunocompetent mouse cancer model is more suitable for evaluating the effectiveness of radiotherapy.
Radiation delivered at low doses, as used in this study, mainly induces mitotic cell death, meaning that cancer cells die during the mitosis which can occur during the first mitosis after treatment and up to 5 cell divisions later. Therefore, only the colony formation assay can adequately determine the survival of cells after irradiation. The trypan blue assay used in this study assesses the integrity of the cell membrane integrity and is not suitable for the purpose of this study. Nevertheless, their in vitro cell survival results show that trastuzumab can increase cancer cell death in combination to radiation therapy. These results should be confirmed with a colony formation assay.
Lines 45 and 76: Radiation therapy mainly creates ROS by ionizing water molecules, not oxygen as mentioned by the authors. Hypoxia also can not prevent the formation of ROS. A better understanding on how ionizing radiation produces free radicals and how it damages DNA in the present and in the absence of oxygen would be appropriate. The authors should also be aware that fractionated radiotherapy allows the redistribution of oxygen through the tumors during the treatment. Therefore, oxygen concentration in a hypoxic area can increase significantly during the treatment.
Minor comments
The figure 2D has not been commented in the text.
The authors are invited to discuss the reasons why trastuzumab did not increase the level of DNA damage when cancer cells were irradiated with a dose of 6 Gy.
What is the rational for analysing the cytokines IL-2 and IFNɤ.
The quality of the writing of the texts and the analysis of the results could be improved
Round 2
Reviewer 1 Report
Authors did not address all comments. For example
- Need a table for genomic characteristics all cell lines
- Need data from trastuzumab or lapatinib resistant cell lines
- Need phos-gamma-H2X data (Western blot)
- Would like to see the hypoxia-induced HIF Western blot data
- would like to see IHC data of CD8+ and CD4+ T cells infiltration inside the tumor compartment
Reviewer 2 Report
The authors have done a very good job of responding to reviewer's comments and improving the quality of their publication. Last suggestion. The role of B and T lymphocytes as well as Natural Killer cells could be investigated in a syngeneic model of breast cancer by depleting these cells by injecting specific antibodies.
